# MULTI-TASK LEARNING WITH DEEP MODEL BASED REINFORCEMENT LEARNING

**Asier Mujika**
ETH Zürich
Zürich, Switzerland
`asierm@student.ethz.ch`

## ABSTRACT

In recent years, model-free methods that use deep learning have achieved great success in many different reinforcement learning environments. Most successful approaches focus on solving a single task, while multi-task reinforcement learning remains an open problem. In this paper, we present a model based approach to deep reinforcement learning which we use to solve different tasks simultaneously. We show that our approach not only does not degrade but actually benefits from learning multiple tasks. For our model, we also present a new kind of recurrent neural network inspired by residual networks that decouples memory from computation allowing to model complex environments that do not require lots of memory. The code will be released before ICLR 2017.

## 1 INTRODUCTION

Recently, there has been a lot of success in applying neural networks to reinforcement learning, achieving super-human performance in many ATARI games (Mnih et al. (2015); Mnih et al. (2016)). Most of these algorithms are based on $Q$-learning, which is a model free approach to reinforcement learning. This approaches learn which actions to perform in each situation, but do not learn an explicit model of the environment. Apart from that, learning to play multiple games simultaneously remains an open problem as these approaches heavily degrade when increasing the number of tasks to learn.

In contrast, we present a model based approach that can learn multiple tasks simultaneously. The idea of learning predictive models has been previously proposed (Schmidhuber (2015); Santana & Hotz (2016)), but all of them focus on learning the predictive models in an unsupervised way. We propose using the reward as a means to learn a representation that captures only that which is important for the game. This also allows us to do the training in a fully supervised way. In the experiments, we show that our approach can surpass human performance simultaneously on three different games. In fact, we show that transfer learning occurs and it benefits from learning multiple tasks simultaneously.

In this paper, we first discuss why $Q$-learning fails to learn multiple tasks and what are its drawbacks. Then, we present our approach, Predictive Reinforcement Learning, as an alternative to overcome those weaknesses. In order to implement our model, we present a recurrent neural network architecture based on residual nets that is specially well suited for our task. Finally, we discuss our experimental results on several ATARI games.

## 2 PREVIOUS WORK: DEEP $Q$-LEARNING

In recent years, approaches that use Deep $Q$-learning have achieved great success, making an important breakthrough when Mnih et al. (2015) presented a neural network architecture that was able to achieve human performance on many different ATARI games, using just the pixels in the screen as input.

As the name indicates, this approach revolves around the $Q$-function. Given a state $s$ and an action $a$, $Q(s, a)$ returns the expected future reward we will get if we perform action $a$ in state $s$. Formally, the $Q$-function is defined in equation 1.

$$Q(s, a) = \mathbb{E}_{s'} \left[ r + \gamma \max_{a'} Q(s', a') | s, a \right] \qquad (1)$$

For the rest of this subsection, we assume the reader is already familiar with Deep $Q$-learning and we discuss its main problems. Otherwise, we recommend skipping to the next section directly as none of the ideas discussed here are necessary to understand our model.

As the true value of the $Q$-function is not known, the idea of Deep $Q$-learning is iteratively approximating this function using a neural network[1] which introduces several problems.

First, the $Q$-values depend on the strategy the network is playing. Thus, the target output for the network given a state-action pair is not constant, since it changes as the network learns. This means that apart from learning an strategy, the network also needs to remember which strategy it is playing. This is one of the main problems when learning multiple tasks, as the networks needs to remember how it is acting on each of the different tasks. Rusu et al. (2015) and Parisotto et al. (2015) have managed to successfully learn multiple tasks using $Q$-learning. Both approaches follow a similar idea: an expert network learns to play a single game, while a multi-tasking network learns to copy the behavior of an expert for each different game. This means that the multi-tasking network does not iteratively approximate the $Q$-function, it just learns to copy the function that the single-task expert has approximated. That is why their approach works, they manage to avoid the problem of simultaneously approximating all the $Q$-functions, as this is done by each single task expert.

Apart from that, the network has to change the strategy very slightly at each update as drastically changing the strategy would change the $Q$-values a lot and cause the approximation process to diverge/slow-down. This forces the model to interact many times with the environment in order to find good strategies. This is not problematic in simulated environments like ATARI games where the simulation can easily be speed up using more computing power. Still, in real world environments, like for example robotics, this is not the case and data efficiency can be an important issue.

## 3    PREDICTIVE REINFORCEMENT LEARNING

In order to avoid the drawbacks of Deep $Q$-learning, we present Predictive Reinforcement Learning (PRL). In our approach, we separate the understanding of the environment from the strategy. This has the advantage of being able to learn from different strategies simultaneously while also being able to play strategies that are completely different to the ones that it learns from. We will also argue that this approach makes generalization easier. But before we present it, we need to define what we want to solve.

### 3.1    PREDICTION PROBLEM

The problem we want to solve is the following: given the current state of the environment and the actions we will make in the future, how is our score going to change through time?

To formalize this problem we introduce the following notation:

- $a_i$: The observation of the environment at time $i$. In the case of ATARI games, this corresponds to the pixels of the screen.
- $r_i$: The total accumulated reward at time $i$. In the case of ATARI games, this corresponds to the in-game score.
- $c_i$: The control that was performed at time $i$. In the case of ATARI games, this corresponds to the inputs of the ATARI controller: *up*, *right*, *shoot*, etc.

---

[1]We do not explain the process, but Mnih et al. (2015) give a good explanation on how this is done.

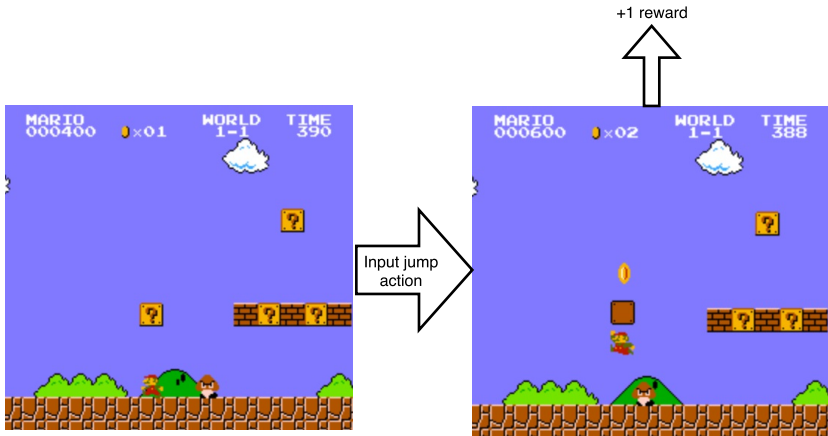

Figure 1: We chose $i = 0$ and $k = 1$. We assume $a_0$ to be the pixels in the current image (the left one) and $c_1$ to be the jump action. Then, given that input, we want to predict $r_1 - r_0$, which is 1, because we earn a reward from time 0 to time 1.

Then, we want to solve the following problem: For a given time $i$ and a positive integer $k$, let the input to our model be an observation $a_i$ and a set of future controls $c_{i+1}, \ldots c_{i+k}$. Then, we want to predict the change in score for the next $k$ time steps, i.e. $(r_{i+1} - r_i), \ldots, (r_{i+k} - r_i)$. Figure 1 illustrates this with an example.

Observe that, unlike in $Q$-learning, our predictions do not depend on the strategy being played. The outputs only depend on the environment we are trying to predict. So, the output for a given state-actions pair is always the same or, in the case of non-deterministic environments, it comes from the same distribution.

## 3.2 MODEL

We have defined what we want to solve but we still need to specify how to implement a model that will do it. We will use neural networks for this and we will divide it into three different networks as follows:

- Perception: This network reads a state $a_i$ and converts it to a lower dimensional vector $h_0$ that is used by the Prediction.

- Prediction: For each $j \in \{1, \ldots, k\}$, this network reads the vector $h_{j-1}$ and the corresponding control $c_{i+j}$ and generates a vector $h_j$ that will be used in the next steps of the Prediction and Valuation. Observe that this is actually a recurrent neural network.

- Valuation: For each $j \in \{1, \ldots, k\}$, this network reads the current vector $h_j$ of the Prediction and predicts the difference in score between the initial time and the current one, i.e, $r_{i+j} - r_i$.

Figure 2 illustrates the model. Observe that what we actually want to solve is a supervised learning problem. Thus, the whole model can be jointly trained with simple backpropagation. We will now proceed to explain each of the components in more detail.

### 3.2.1 PERCEPTION

The Perception has to be tailored for the kind of observations the environment returns. For now, we will focus only on vision based Perception. As we said before, the idea of this network is to convert the high dimensional input to a low dimensional vector that contains only the necessary information for predicting the score. In the case of video games, it is easy to see that such vector exists. The input will consists of thousands of pixels but all we care about is the position of a few key objects, like for example, the main character or the enemies. This information can easily be

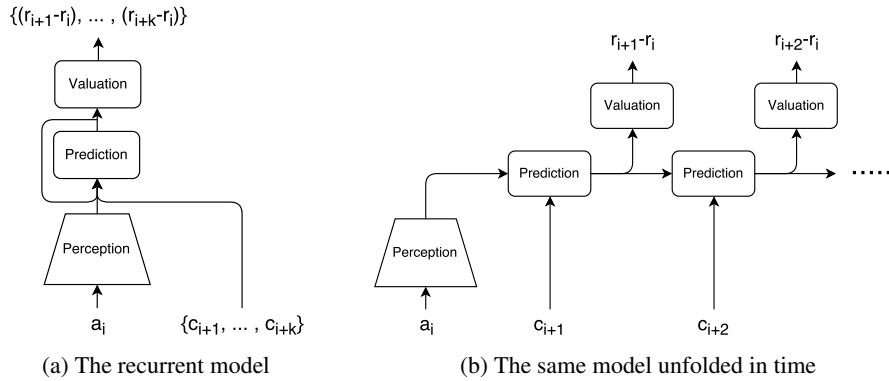

(a) The recurrent model (b) The same model unfolded in time

Figure 2: Diagram of our predictive model.

encoded using very few neurons. In our experiments, we convert an input consisting of $28K$ pixels into a vector of just $100$ real values.

In order to do this, we use deep convolutional networks. These networks have recently achieved super-human performance in very complex image recognition tasks (He et al., 2015). In fact, it has been observed that the upper layers in these models learn lower dimensional abstract representations of the input (Yosinski et al. (2015), Karpathy & Li (2015)). Given this, it seems reasonable to believe that if we use any of the successful architectures for vision, our model will be able to learn a useful representation that can be used by the Prediction.

### 3.2.2 PREDICTION

For the Prediction network, we present a new kind of recurrent network based on residual neural networks (He et al., 2015), which is specially well suited for our task and it achieved better results than an LSTM (Hochreiter & Schmidhuber, 1997) with a similar number of parameters in our initial tests.

**Residual Recurrent Neural Network (RRNN)** We define the RRNN in Figure 3 using the following notation: $LN$ is the layer normalization function (Ba et al., 2016) which normalizes the activations to have a median of $0$ and standard deviation of $1$. "$\cdot$" is the concatenation of two vectors. $f$ can be any parameterizable and differentiable function, e.g., a multilayer perceptron.

$$r_i = f(LN(h_{i-1}) \cdot x_i) \quad (2)$$
$$h_i = h_{i-1} + r_i \quad (3)$$

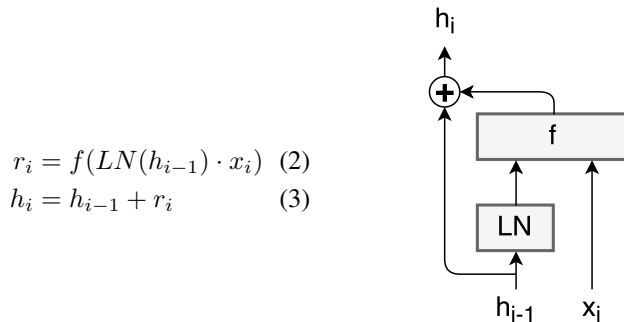

Figure 3: The equations of the RRNN and a diagram of the network.

As in residual networks, instead of calculating what the new state of the network should be, we calculate how it should change ($r_i$). As shown by He et al. (2015) this prevents vanishing gradients or optimization difficulties. $LN$ outputs a vector with mean $0$ and standard deviation $1$. As we

proof[2] in Observation 1, this prevents internal exploding values that may arise from repeatedly adding $r$ to $h$. It also avoids the problem of vanishing gradients in saturating functions like sigmoid or hyperbolic tangent.

**Observation 1.** Let $x \in \mathbb{R}^n$ be a vector with median 0 and standard deviation 1. Then, for all $1 \leq i \leq n$, we get that $x_i \leq \sqrt{n}$.

*Proof.* Taking into account that the median is 0 and the standard deviation is 1, simply substituting the values in the formula for the standard deviation shows the observation.

$$\sigma = \sqrt{\frac{1}{n}\sum_{j=1}^{n}(x_j - \mu)^2} \tag{4}$$

$$1 = \sqrt{\frac{1}{n}\sum_{j=1}^{n}x_j^2} \tag{5}$$

$$\sqrt{n} = \sqrt{\sum_{j=1}^{n}x_j^2} \tag{6}$$

$$\sqrt{n} \geq x_i \tag{7}$$

$\square$

The idea behind this network is mimicking how a video game's logic works. A game has some variables (like positions or speeds of different objects) that are slightly modified at each step. Our intuition is that the network can learn a representation of these variables ($h$), while $f$ learns how they are transformed at each frame. Apart from that, this model decouples memory from computation allowing to increase the complexity of $f$ without having to increase the number of neurons in $h$. This is specially useful as the number of real valued neurons needed to represent the state of a game is quite small. Still, the function to move from one frame to the next can be quite complex, as it has to model all the interactions between the objects such as collisions, movements, etc.

Even if this method looks like it may be just tailored for video games, it should work equally well for real world environments. After all, physics simulations that model the real world work in the same way, with some variables that represent the current state of the system and some equations that define how that system evolves over time.

### 3.2.3 VALUATION

The Valuation network reads the $h$ vector at time $i + j$ and outputs the change in reward for that time step, i.e. $r_{i+j} - r_j$. Still, it is a key part of our model as it allows to decouple the representation learned by the Prediction from the reward function. For example, consider a robot in a real world environment. If the Perception learns to capture the physical properties of all surrounding objects (shape, mass, speed, etc.) and the Prediction learns to make a physical simulation of the environment, this model can be used for any possible task in that environment, only the Valuation would need to be changed.

### 3.3 STRATEGY

As we previously said, finding an optimal strategy is a very hard problem and this part is the most complicated. So, in order to test our model in the experiments, we opted for hard-coding a strategy. There, we generate a set of future controls uniformly at random and then we pick the one that would maximize our reward, given that the probability of dying is low enough. Because of this, the games we have tried have been carefully selected such that they do not need very sophisticated and long-term strategies.

---

[2]The bound is not tight but it is sufficient for our purposes and straightforward to prove.

|            | Input   | Output |
|------------|---------|--------|
| ReLU + Linear | 100 + 3 | 500 |
| ReLU + Linear | 500     | 100 |

Table 1: $f$ function of the Prediction network. We apply the non-linearity before the linear layer, this way we avoid always adding positive values. The ReLU is not applied to the control inputs.

|               | Input | Output |
|---------------|-------|--------|
| LN            | 100   | 100 |
| Linear + ReLU | 100   | 100 |
| Linear + Sigmoid | 100 | 2 |

Table 2: Valuation network. We apply Layer Normalization to bound the incoming values to the network.

Still, our approach learns a predictive model that is independent of any strategy and this can be beneficial in two ways. First, the model can play a strategy that is completely different to the ones it learns from. Apart from that, learning a predictive model is a very hard task to over-fit. Consider a game with 10 possible control inputs and a training set where we consider the next 25 time steps. Then, there are $10^{25}$ possible control sequences. This means that every sequence we train on is unique and this forces the model to generalize. Unfortunately, there is also a downside. Our approach is not able to learn from good strategies because we test our model with many different ones in order to pick the best. Some of these strategies will be quite bad and thus, the model needs to learn what makes the difference between a good and a bad set of moves.

## 4 EXPERIMENTS

### 4.1 ENVIRONMENT

Our experiments have been performed on a computer with a GeForce GTX 980 GPU and an Intel Xeon E5-2630 CPU. For the neural network, we have used the Torch7 framework and for the ATARI simulations, we have used Alewrap, which is a Lua wrapper for the Arcade Learning Environment (Bellemare et al., 2015).

### 4.2 MODEL

For the Perception, we used a network inspired in deep residual networks (He et al., 2015). Figure 4 shows the architecture. The reason for this, is that even if the Perception is relatively shallow, when unfolding the Prediction network over time, the depth of the resulting model is over 50 layers deep.

For the Prediction, we use a Residual Recurrent Neural Network. Table 1 describes the network used for the $f$ function. Finally, Table 2 illustrates the Valuation network.

### 4.3 SETUP

In our experiments, we have trained on three different ATARI games simultaneously: Breakout, Pong and Demon Attack.

We preprocess the images following the same technique of Mnih et al. (2015). We take the maximum from the last 2 frames to get a single $84 \times 84$ black and white image for the current observation. The input to the Perception is a $4 \times 84 \times 84$ tensor containing the last 4 observations. This is necessary to be able to use a feed-forward network for the Perception. If we observed a single frame,

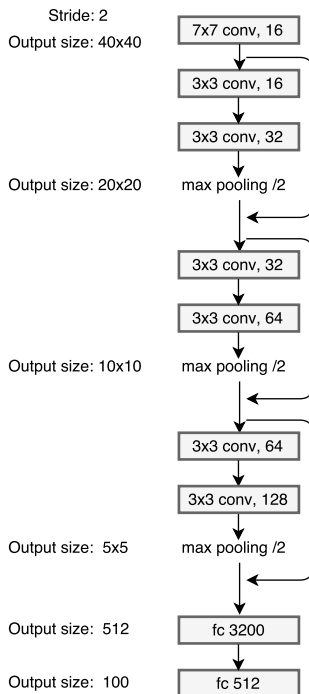

Figure 4: Each layer is followed by a Batch Normalization (Ioffe & Szegedy, 2015) and a Rectifier Linear Unit.

it would not be possible to infer the speed and direction of a moving object. Not doing this would force us to use a recurrent network on the Perception, making the training of the whole model much slower.

In order to train the Prediction, we unfold the network over time (25 time steps) and treat the model as a feed-forward network with shared weights. This corresponds to approximately 1.7 seconds.

For our Valuation, network we output two values. First, the probability that our score is higher than in the initial time step. Second, we output the probability of dying. This is trained using cross entropy loss.

To train the model, we use an off-line learning approach for simplicity. During training we alternate between two steps. First, generate and store data and then, train the model off-line on that data.

## 4.4 GENERATING DATA

In order to generate the data, we store tuples $(a_i, C = \{c_{i+1}, \ldots c_{i+25}\}, R = \{r_{i+1} - r_i, \ldots r_{i+25} - r_i\})$ as we are playing the game. That is, for each time $i$, we store the following:

- $a_i$: A $4 \times 84 \times 84$ tensor, containing 4 consecutive black and white frames of size $84 \times 84$ each.

- $C$: For $j \in \{i+1, \ldots, i+25\}$, each $c_j$ is a 3 dimensional vector that encodes the control action performed at time $j$. The first dimension corresponds to the *shoot* action, the second to horizontal actions and the third to vertical actions. For example, $[1, -1, 0]$ represent pressing *shoot* and *left*.

- $R$: For $j \in \{i+1, \ldots, i+25\}$, we store a 2 dimensional binary vector $r_j$. $r_{j1}$ is 1 if we die between time $i$ and $j$. $r_{j2}$ is 1 if we have not lost a life and we also earn a point between time $i$ and $j$.

Initially, we have an untrained model, so at each time step, we pick an action uniformly at random and perform it. For the next iterations, we pick a $k$ and do the following to play the game:

1. Run the Perception network on the last 4 frames to obtain the initial vector.

2. Generate $k - 1$ sequences of 25 actions uniformly at random. Apart from that, take the best sequence from the previous time step and also consider it. This gives a total of $k$ sequences. Then, for each sequence, run the Prediction and Valuation networks with the vector obtained in Step 1.

3. Finally, pick a sequence of actions as follows. Consider only the moves that have a low enough probability of dying. From those, pick the one that has the highest probability of earning a point. If none has a high enough probability, just pick the one with the lowest probability of dying.

We start with $k = 25$ and increase it every few iterations up to $k = 200$. For the full details check Appendix A. In order to accelerate training, we run several games in parallel. This allows to run the Perception, Prediction and Valuation networks together with the ATARI simulation in parallel, which heavily speeds up the generation of data without any drawback.

## 4.5 TRAINING

In the beginning, we generate $400K$ training cases for each of the games by playing randomly, which gives us a total of $1.2M$ training cases. Then, for the subsequent iterations, we generate $200K$ additional training cases per game ($600K$ in total) and train again on the whole dataset. That is, at first we have $1.2M$ training cases, afterwards $1.8M$, then $2.4M$ and so on.

|  | Pong | Breakout | Demon |
|---|---|---|---|
| Human score | 9.3 | 31.8 | 3401 |
| Random | -20.7 | 1.7 | 152 |
| PRL Iteration 2 | -8.62 | 13.2 | 1220 |

Table 3: After one iteration Preditive Reinforcement Learning (PRL) has only observed random play but it can play much better. This means that it is able to generalize well to many situations it has not observed during training.

|  | Pong | Breakout | Demon |
|---|---|---|---|
| Human score | 9.3 | 31.8 | 3401 |
| PRL Best (Multi-task) | 14.6 | 316 | 6872 |
| PRL Best (Single-task) | 18.2 | 186 | 6100 |
| A3C (Mnih et al., 2016) | 18.9 | 766.8 | 115202 |

Table 4: The best iteration of PRL is able to surpass human performance in all three tasks. Still, state of the art model-free approaches work better.

The training is done in a supervised way as depicted in Figure 2b. $a_i$ and $C$ are given as input to the network and $R$ as target. We minimize the cross-entropy loss using mini-batch gradient descent. For the full details on the learning schedule check Appendix A.

In order to accelerate the process, instead of training a new network in each iteration, we keep training the model from the previous iteration. This has the effect that we would train much more on the initial training cases while the most recent ones would have an ever smaller effect as the training set grows. To avoid this, we assign a weight to each iteration and sample according to these weights during training. Every three iterations, we multiply by three the weights we assign to them. By doing this, we manage to focus on recent training cases, while still preserving the whole training set.

Observe that we never tell our network which game it is playing, but it learns to infer it from the observation $a_i$. Also, at each iteration, we add cases that are generated using a different neural network. So our training set contains instances generated using many different strategies.

## 4.6 RESULTS

We have trained a model on the three games for a total of 19 iterations, which correspond to $4M$ time steps per game (74 hours of play at 60 Hz). Each iteration takes around two hours on our hardware. We have also trained an individual model for each game for $4M$ time steps. In the individual models, we reduced the length of the training such that the number of parameter updates per game is the same as in the multi-task case. Unless some kind of transfer learning occurs, one would expect some degradation in performance in the multi-task model. Figure 5 shows that not only there is no degradation in Pong and Demon Attack, but also that there is a considerable improvement in Breakout. This confirms our initial belief that our approach is specially well suited for multi-task learning.

We have also argued that our model can potentially play a very different strategy from the one it has observed. Table 3 shows that this is actually the case. A model that has learned only from random play is able to play at least 7 times better.

Demon Attack's plot in Figure 5c shows a potential problem we mentioned earlier which also happens in the other two games to a lesser extent. Once the strategy is good enough, the agent dies very rarely. This causes the model to "forget" which actions lead to a death and makes the score oscillate.

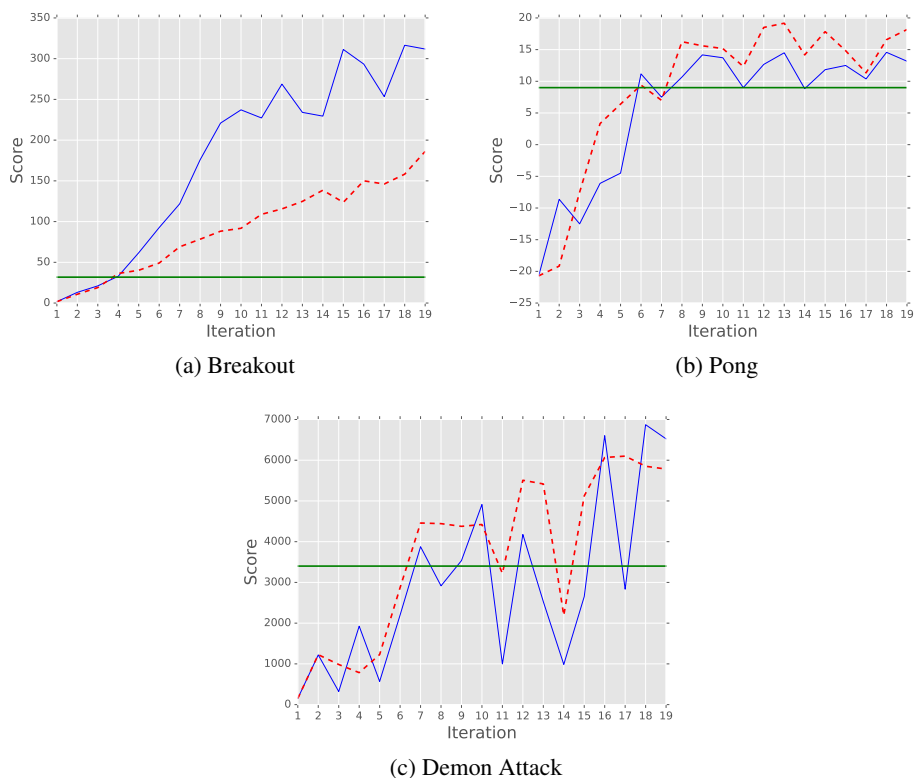

(a) Breakout

(b) Pong

(c) Demon Attack

Figure 5: Comparison between an agent that learns the three games simultaneously (continuous blue), one that learns each game individually (dashed red) and the score of human testers (horizontal green) as reported by Mnih et al. (2015).

## 5 DISCUSSION

We have presented a novel model based approach to deep reinforcement learning. Despite not achieving state of the art results, this papers opens new lines of research showing that a model based approach can work in environments as complex as ATARI. We have also shown that it can beat human performance in three different tasks simultaneously and that it can benefit from learning multiple tasks.

Still, the model has two areas that can be addressed in future work: long-term dependencies and the instability during training. The first, can potentially be solved by combining our approach with $Q$-learning based techniques. For the instability, balancing the training set or oversampling hard training cases could alleviate the problem.

Finally, we have also presented a new kind of recurrent network which can be very useful for problems were little memory and a lot of computation is needed.

ACKNOWLEDGMENTS

I thank Angelika Steger and Florian Meier for their hardware support in the final experiments and comments on previous versions of the paper.

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

# Appendices

## A IMPLEMENTATION DETAILS

Due to the huge cost involved in training the agents, we have not exhaustively searched over all the possible hyper parameters. Still, we present them here for reproducibility of the results.

- *Number of strategies*: As explained in Section 4.4, we need to pick a number $k$ of strategies we consider at each step. Initially, we pick $k = 25$, raise it to $k = 100$ at iteration 4 and finally, at iteration 7, we set it to $k = 200$ for the remaining of the experiment.

- *Confidence interval*: We also need to pick how safe we want to play, i.e., where we set the threshold for the set of actions we consider. For simplicity, in Breakout and Pong, we set it to 0 and only pick the safest option. In Demon Attack, initially we only consider actions with a survival probability higher than 0.2 for three iterations. After that, we reduce it to 0.1 for another three iterations. Then, we set it to 0.005 until iteration 15 and finally, reduce it to 0.001 for the rest of the iterations.

- *Learning schedule*: For training we use the Adam (Kingma & Ba, 2014) optimizer with a batch size of 100. We use a learning rate of $10^{-4}$ for the first 3 iterations, then reduce it to $5 \times 10^{-5}$ for the next 3 iterations and finally set it to $10^{-5}$ for the rest of the experiment. We make a total of $4.8 \times 10^4$ parameter updates per iteration ($1.6 \times 10^4$ in the case of single-task networks) and divide the learning rate in half after $2.4 \times 10^4$ updates for the remaining of the iteration. We add a weight decay of 0.0001 and clamp the gradients element-wise to the $[-1, 1]$ range.

Apart from that, at the beginning of each episode, we pick an $n \in [0, 30]$ uniformly at random and do not perform any action for the initial $n$ time steps of that episode. This idea was also used by Mnih et al. (2015) to avoid any possible over-fitting. In addition, we also press *shoot* to start a new episode every time we die in Breakout, since in the first iterations the model learns that the safest option is not to start a new episode. This causes the agent to waste a lot of time without starting a new episode.

