# Peer review of "Multi-task learning with deep model based reinforcement learning"

_ICLR 2017 — rejected_

[Official Review · AnonReviewer3 · rating 4 · confidence 4 · 15 Dec 2016]
**Encouraging results but the approach is too ad-hoc**

This paper proposes a new approach to model based reinforcement learning and
evaluates it on 3 ATARI games. The approach involves training a model that
predicts a sequence of rewards and probabilities of losing a life given a
context of frames and a sequence of actions. The controller samples random
sequences of actions and executes the one that balances the probabilities of
earning a point and losing a life given some thresholds. The proposed system
learns to play 3 Atari games both individually and when trained on all 3 in a
multi-task setup at super-human level.

The results presented in the paper are very encouraging but there are many
ad-hoc design choices in the design of the system. The paper also provides
little insight into the importance of the different components of the system.

Main concerns:
- The way predicted rewards and life loss probabilities are combined is very ad-hoc.
  The natural way to do this would be by learning a Q-value, instead different
  rules are devised for different games.
- Is a model actually being learned and improved? It would be good to see
  predictions for several actions sequences from some carefully chosen start
  states. This would be good to see both on a game where the approach works and
  on a game where it fails. The learning progress could also be measured by
  plotting the training loss on a fixed holdout set of sequences.
- How important is the proposed RRNN architecture? Would it still work without
  the residual connections? Would a standard LSTM also work?

Minor points:
- Intro, paragraph 2 - There is a lot of much earlier work on using models in
  RL. For example, see Dyna and "Memory approaches to reinforcement learning in
  non-Markovian domains" by Lin and Mitchell to name just two.
- Section 3.1 - Minor point, but using a_i to represent the observation is
  unusual.  Why not use o_i for observations and a_i for actions?
- Section 3.2.2 - Notation again, r_i was used earlier to represent the
  reward at time i but it is being used again for something else.
- Observation 1 seems somewhat out of place. Citing the layer normalization
  paper for the motivation is enough.
- Section 3.2.2, second last paragraph - How is memory decoupled from
  computation here? Models like neural turning machines accomplish this by using
  an external memory, but this looks like an RNN with skip connections.
- Section 3.3, second paragraph - Whether the model overfits or not depends on
  the data. The approach doesn't work with demonstrations precisely because it
  would overfit.
- Figure 4 - The reference for Batch Normalization should be Ioffe and Szegedy
  instead of Morimoto et al.

Overall I think the paper has some really promising ideas and encouraging
results but is missing a few exploratory/ablation experiments and some polish.

[Official Review · AnonReviewer1 · rating 2 · confidence 5 · 16 Dec 2016 (modified: 23 Jan 2017)]

The term strategy is a bit ambiguous. Could you please explain more in formal terms what is strategy?
Is r the discounted Return at time t, or the reward at time t?
Could the author compare the method to TD learning?
The paper is vague and using many RL terms with different meanings without clarifying those diversions.
"So, the output for a given state-actions pair is always same". Q function by definition is the value of (state, action). So as long as the policy is deterministic the output would be always same too. How's this different from Q learning?
The model description doesn't specify what is the policy, and it's only being mentioned in data generation part.
Why is it a model based approach?
The learning curves are only for 19 iterations, which does not give any useful information. The final results are clearly nothing comparable to previous works. The model is only being tested on three games.

The paper is vague and using informal language or sometimes misusing the common RL terms. The experiments are very small scale and even in that scenario performing very bad. It's not clear, why it's a model-based approach.

[Official Review · AnonReviewer2 · rating 4 · confidence 4 · 17 Dec 2016]
**Interesting ideas but investigated too superficially**

This paper proposes a model-based reinforcement learning approach focusing on predicting future rewards given a current state and future actions. This is achieved with a "residual recurrent neural network", that outputs the expected reward increase at various time steps in the future. To demonstrate the usefulness of this approach, experiments are conducted on Atari games, with a simple playing strategy that consists in evaluating random sequences of moves and picking the one with highest expected reward (and low enough chance of dying). Interestingly, out of the 3 games tested, one of them exhibits better performance when the agent is trained in a multitask setting (i.e. learning all games simultaneously), hinting that transfer learning is occurring.

This submission is easy enough to read, and the reward prediction architecture looks like an original and sound idea. There are however several points that I believe prevent this work from reaching the ICLR bar, as detailed below.

The first issue is the discrepancy between the algorithm proposed in Section 3 vs its actual implementation in Section 4 (experiments): in Section 3 the output is supposed to be the expected accumulated reward in future time steps (as a single scalar), while in experiments it is instead two numbers, one which is the probability of dying and another one which is the probability of having a higher score without dying. This might work better, but it also means the idea as presented in the main body of the paper is not actually evaluated (and I guess it would not work well, as otherwise why implement it differently?)

In addition, the experimental results are quite limited: only on 3 games that were hand-picked to be easy enough, and no comparison to other RL techniques (DQN & friends). I realize that the main focus of the paper is not about exhibiting state-of-the-art results, since the policy being used is only a simple heuristic to show that the model predictions can ne used to drive decisions. That being said, I think experiments should have tried to demonstrate how to use this model to obtain better reinforcement learning algorithms: there is actually no reinforcement learning done here, since the model is a supervised algorithm, used in a manually-defined hardcoded policy. Another question that could have been addressed (but was not) in the experiments is how good these predictions are (e.g. classification error on dying probability, MSE on future rewards, ...), compared to simpler baselines.

Finally, the paper's "previous work" section is too limited, focusing only on DQN and in particular saying very little on the topic of model-based RL. I think a paper like for instance "Action-Conditional Video Prediction using Deep Networks in Atari Games" should have been an obvious "must cite".

Minor comments:
- Notations are unusual, with "a" denoting a state rather than an action, this is potentially confusing and I see no reason to stray away from standard RL notations
- Using a dot for tensor concatenation is not a great choice either, since the dot usually indicates a dot product
- The r_i in 3.2.2 is a residual that has nothing to do with r_i the reward
- c_i is defined as "The control that was performed at time i", but instead it seems to be the control performed at time i-1
- There is a recurrent confusion between mean and median in 3.2.2
- x should not be used in Observation 1 since the x from Fig. 3 does not go through layer normalization
- The inequality in Observation 1 should be about |x_i|, not x_i
- Observation 1 (with its proof) takes too much space for such a simple result
- In 3.2.3 the first r_j should be r_i
- The probability of dying comes out of nowhere in 3.3, since we do not know yet it will be an output of the model
- "Our approach is not able to learn from good strategies" => did you mean "*only* from good strategies"?
- Please say that in Fig. 4 "fc" means "fully connected"
- It would be nice also to say how the architecture of Fig. 4 differs from the classical DQN architecture from Mnih et al (2015)
- Please clarify r_j2 as per your answer in OpenReview comments
- Table 3 says "After one iteration" but has "PRL Iteration 2" in it, which is confusing
- "Figure 5 shows that not only there is no degradation in Pong and Demon Attack"=> to me it seems to be a bit worse, actually
- "A model that has learned only from random play is able to play at least 7 times better." => not clear where this 7 comes from
- "Demon Attack's plot in Figure 5c shows a potential problem we mentioned earlier" => where was it mentioned?

[Final Decision · Program Chairs · 06 Feb 2017]
**ICLR committee final decision**

The authors have proposed a new method for deep RL that uses model-based evaluation of states and actions and reward/life loss predictions. The evaluation, on just 3 ATari games with no comparisons to state of the art methods, is insufficient, and the method seems ad-hoc and unclear. Design choices are not clearly described or justified. The paper gives no insight as to how the different aspects of the approach relate or contribute to the overall results.